# Effectiveness of Fascia Iliaca Compartment Block after Elective Total Hip Replacement: A Prospective, Randomized, Controlled Study

**DOI:** 10.3390/ijerph18094891

**Published:** 2021-05-04

**Authors:** Wojciech Gola, Szymon Bialka, Aleksander J. Owczarek, Hanna Misiolek

**Affiliations:** 1Department of Anaesthesia and Intensive Care, Saint Lucas Hospital, Konskie 26-200, Poland; 2Faculty of Medicine and Health Sciences, Jan Kochanowski University, Kielce 25-369, Poland; 3Intensive Care and Emergency Medicine, Department of Anaesthesiology, Faculty of Medical Sciences in Zabrze, Medical University of Silesia, Katowice 40-055, Poland; hmisiolek@sum.edu.pl; 4Health Promotion and Obesity Management Unit, Department of Pathophysiology, Medical Faculty in Katowice, Medical University of Silesiain, Katowice 40-055, Poland; aowczarek@sum.edu.pl

**Keywords:** regional anesthesia, multimodal analgesia, fascia iliaca compartment block, total hip replacement

## Abstract

Objective: An assessment of the feasibility of fascia iliaca compartment block (FICB) combined with nonopioid analgesics and patient controlled analgesia (PCA), oxycodone, in the perioperative anaesthetic management for elective total hip replacement (THR). Design: A randomised, single-center, open-label study. Setting: A single hospital. The study was conducted from October 2018 to May 2019. Participants: In total, 109 patients were scheduled for elective total hip replacement. Interventions: Postoperative FICB with 0.375% ropivacaine in conjunction with nonopioid analgesics (paracetamol, metamizole, and pregabalin) and oxycodone as rescue analgesia. Measurements: Pain intensity was measured using the Numeric Pain Rating Scale (NRS) at rest and during rehabilitation, the total dose of postoperative oxycodone required, the occurrence of opioid-related adverse events, patient hospitalisation time, and level of satisfaction. Follow-up period: 48 h. Main Results: A total of 109 patients were randomised into two groups and, of these, 9 were subsequently excluded from the analysis (three conversions to general anaesthesia, two failures to perform FICB, four failures to use the PCA pump). Patients in the FICB group received standard intravenous analgesia with FICB, and those in the control group were managed with standard intravenous analgesia only. Pain level measured with NRS was significantly lower at rest and during rehabilitation in the FICB group. Oxycodone use in the first 48 h was significantly higher in the control group (*p* < 0.001); additionally, the time to the first dose of rescue analgesia was significantly shorter (*p* < 0.001). In the control group, there was a higher rate of side effects and a significantly longer hospitalisation time (*p* < 0.001). Similarly, higher satisfaction with the applied analgesic treatment was noted in the FICB group. Conclusions: FICB in elective THR treatments is an effective form of analgesia, which reduces the need for opioids, the number of complications, the length of hospitalisation, and which ensures a high level of patient satisfaction with the analgesic treatment used. Trial registration: ClinicalTrials.gov No. NCT04690647.

## 1. Introduction

The development of joint reconstructive surgery is considered to be one of the most important advances in interventional orthopaedics in recent years [1,2]. Total hip replacement (THR) is a treatment of choice for the advanced osteoarthritis of the hip joint [1,3,4]. The number of THR procedures is constantly growing due to the increasingly broader indications and social awareness, as well as a longer professional activity and improved access to medical care. At the same time, THR is the most common hip reconstructive surgery and one of the most common orthopaedic procedures globally [5,6].

One of the fundamental elements of modern hip reconstructive surgery is to ensure optimal analgesia while minimising the need for opioids and reducing adverse effects associated with their systemic administration. The most common adverse reactions include postoperative nausea and vomiting, oversedation, apnoea, and respiratory complications. Early rehabilitation and efficient mobilisation of the patient is another element of perioperative care in joint replacement surgeries [7,8].

Regional anaesthetic techniques and local infiltration of the surgical site using local anaesthetics (LAs) are the primary elements of modern, multimodal treatment approach for acute postoperative pain, contributing to both improved control of postoperative pain and early patient mobilisation and rehabilitation [9,10,11,12]. Despite a large number of scientific reports, the choice of an optimal form of analgesia, including regional block for elective hip replacement, is still not fully defined. Furthermore, there are no uniform guidelines or strong recommendations [13].

Fascia iliaca compartment block (FICB) is one of the regional nerve blocks used in THR [14,15,16]. Indications for FICB include pre-, peri- and postoperative analgesia after fractured neck of the femur. Additional indications include hip and knee surgery, above-knee amputation, and application of plaster cast to femoral fracture in paediatric patients, although data to support these indications are limited [17].

The main aim of the study was the assessment of FICB efficacy in elective THR. The following parameters were assessed: NRS at rest and during rehabilitation, the need for opioids, adverse effects related to their systemic administration, time to first analgesic intervention, and patient satisfaction with the analgesic regimen.

## 2. Materials and Methods

This was a prospective, observational, randomised study. The study was approved by the Ethics Committee of the District Medical Chamber in Kielce, Poland (protocol code 80/2018, date of approval 27 September 2018) and was registered under Clinical Trial No. NCT04690647. Patients qualified for elective posterolateral hip replacement and hospitalised in the Saint Lucas Hospital in Końskie between October 2018 and May 2019 were included in the study. The inclusion criteria were as follows: age > 18 and < 75 years, American Society of Anaesthesiology (ASA) physical status I–III, BMI 19–30 kg/m^2^, no contraindications for postoperative pharmacotherapy, and analgesia used in the study. Patients with contraindications for spinal anaesthesia and regional blocks, previously diagnosed with chronic pain, chronic use of opioids, obesity (BMI > 30 kg/m^2^), allergy to drugs used in the study, and mental state preventing proper use of patient-controlled analgesia (PCA) pump were excluded from the study.

Patients meeting the inclusion criteria were randomised to one of two groups receiving different postoperative analgesic regimens:Control group—standard intravenous postoperative analgesia,FICB group—standard intravenous postoperative analgesia + ultrasound-guided supra-inguinal fascia iliaca compartment block (S-FICB).

Having been thoroughly informed about the aims and method of the study, and after giving informed consent, the patients were randomly allocated to the control group or the FICB group. The allocation numbers were computer-generated using the www.random.org website.

### 2.1. Preoperative Care and Spinal Anaesthesia

The pre- and intra-operative management protocol was standardised for all patients qualified for the study. No standard premedication was used. All patients received preventive analgesia in the form of oral paracetamol (Paracetamol Biofarm, Poznan, Poland) at 500 mg, oral metamizole (Pyralginum, Polpharma, Starogard Gdanski, Poland) at 500 mg, and oral pregabalin (Lyrica, Pfizer, New York, NY, USA) at 75 mg an hour before the procedure. After arriving at the operating room and starting standard monitoring (including noninvasive blood pressure monitoring, electrocardiogram and pulse oximetry) within the block room, spinal anaesthesia was performed. For this purpose, the patient was placed in a sitting knee-chest position. Then, after surgical disinfection of the puncture site (L2/L3 or L3/L4), the subarachnoid space was identified at midline using a needle for spinal anaesthesia (Pencan 27 G, 0.42 × 50 mm, B.Braun, Melsungen, Germany) under sterile conditions. After clear, colourless cerebrospinal fluid appeared, 1.7–2.2 mL of 0.5% hyperbaric bupivacaine solution (Marcaine Spinal 0.5% Heavy, AstraZeneca, Cambridge, England, UK) was injected. The extent of the sensory block was assessed using an aerosol disinfectant. Once the optimal extent of sensory blockade (up to the level of Th10) was achieved, the patient was transported to the operating room and placed in a lateral position to perform the procedure.

### 2.2. Intraoperative Care

During the surgery, sedation using IV propofol (Propofol 1% MCT/LCT Fresenius, Fresenius Kabi, Bad Homburg, Germany) at 25–50 µg/kg/min or headphones connected to an audio system (relaxing music) were used on patient’s request [18,19]. Standard intraoperative fluid therapy using balanced crystalloids at a dose of 1–2 mL/kg/h was used. In the event of bradycardia (HR < 50 bpm or a decrease >20% of the baseline value), IV atropine (Atropine Sulfuricum, Polfa Warszawa, Poland) was administered in fractionated doses of 0.01 mg/kg of body weight up to a maximum dose of 2 mg. In the case of mean blood pressure drop below 70 mmHg or >25% vs. baseline, IV ephedrine (Ephedrinum Hydrochloricum WZF, Polfa Warszawa, Warsaw, Poland) in fractionated doses of 5 mg (maximum dose 25 mg) or (if ineffective) noradrenaline infusion (Levonor, Polfa Warszawa, Warsaw, Poland) in a syringe pump, titrated to maintain blood pressure >70 mmHg, was used.

### 2.3. Postoperative Care

After the procedure, the patients stayed in the postoperative supervision room and were monitored for vital parameters. After obtaining a minimum score of 9 on the Aldrete’s Scoring System twice, approximately 30 min apart, the patients were discharged to the surgical unit. All patients in the postoperative supervision room additionally received oxycodone (OxyNorm Mundipharma, Basel, Switzerland) in the PCA system with a bolus of 1 mg and a 10-min lock-out. The patients also received an information brochure on the use of the PCA pump and were instructed on how to use the pump. Standard multimodal analgesia with the use of nonopioid and opioid analgesics was used in the study groups [9]. Following the concept of multimodal analgesia, the patients received comprehensive therapy according to the following regimen: In the first 24 h after surgery: IV paracetamol (Paracetamol, B. Braun, Melsungen, Germany) at 1 g every 6 h, IV metamizole at 1 g every 6 h, and oral pregabalin at 75 mg once a day. Oral oxycodone (OxyContin Mundipharma, Basel, Switzerland) at 10 mg every 12 h was included on the first postoperative day in the evening. On day 2 and subsequent postoperative days, the patients received oral paracetamol at 500 mg every 6 h, oral metamizole at 500 mg every 6 h, and oral pregabalin at 75 mg once daily. In the FICB group, an additional ultrasound-guided S-FICB was performed.

### 2.4. Suprainguinal Fascia Iliaca Compartment Block

After the procedure, S-FICB was performed in the postoperative supervision room. The block was performed by anaesthesiology residents under the supervision of a consultant or by the consultant. After anaesthesia site inspection, prescanning, and sterile preparation of the field, a sterile-sheathed linear transducer (L 10–14 MHz, SonoSite M-Turbo, Bothell, WA, USA) was applied along the inguinal sulcus, at the level of femoral vessels and nerve. After initial identification of the most important anatomical structures (femoral vessels, femoral nerve, iliac fascia, iliopsoas muscle), the transducer was placed laterally to the femoral nerve and then rotated 90 degrees to the sagittal plane. Then the transducer was moved cranially along the long axis of the iliopsoas muscle. The optimal position of the transducer is a level above the inguinal ligament, directly medial to the iliac plate, at a site where the iliopsoas muscle passes the iliac plate, running towards the minor pelvis. Then, under ultrasound guidance, a regional block needle (22 G, 50 mm, Stimuplex Ultra, B.Braun, Melsungen, Germany) was introduced in the cranial direction. Once good needle alignment with the ultrasound beam was achieved, the needle was inserted deep into the tissues until an optimal position of the needle tip was obtained. Needle location was additionally verified by injecting 2–3 mL of 0.9% NaCl and observing the spread of solution within the tissues. Once the correct position of the needle tip was confirmed, the local anaesthetic was deposited under the iliac fascia so as to force its flow towards the lumbar plexus. Forty milliliters of 0.375% ropivacaine solution (1% Ropimol Molteni, Florence, Italy) with adrenaline (Adrenaline WZF 0.1%, Polfa Warszawa, Warsaw, Poland) at a dose of 5 μg/mL of solution was used for the block. LA was administered as 5 mL boluses with a 20-s interval, each time preceded with a pre-injection aspiration to avoid intravascular injection. After the block, the patient remained in the postoperative supervision room with full monitoring of vital functions for at least 40 min.

### 2.5. Outcome Measures

The following parameters were recorded in the post-anesthesia care unit (PACU) and in the first 48 h: heart rate (HR/bpm); systolic Blood Pressure (SBP/mmHg); diastolic blood pressure (DBP/mmHg;); mean arterial pressure (MAP/mmHg) measured non-invasively at 4-h intervals; NRS pain severity at 4, 8, 12, 24, and 48 h both at rest and during rehabilitation; time to first analgesic intervention; oxycodone consumption within 48 h; adverse reactions after certain treatment methods/drugs (such as constipation, postoperative nausea and vomiting; PONV); hypotension defined MAP < 70 mmHg; bradycardia defined as HR < 50 bpm; oversedation defined as Richmond Agitation-Sedation Scale (RASS) score of −1 [20]; overall satisfaction with the analgesic treatment used based on the Likert scale [21]; and the length of stay (LOS).

### 2.6. Statistical Analysis

Data in the interval scale with a normal distribution were presented as mean ± standard deviation, and as the median (lower quartile–upper quartile) in the case of data with non-normal or skewed distribution. The normality of the obtained result’s distribution was assessed with the Shapiro-Wilk test and the quantile plot (Q-Q). Nominal and ordinal data are presented as numbers and percentages. The χ2 test was used to compare variables on the nominal and ordinal scales, including dichotomous ones, and the χ2 test with a Yates correction (for two-way tables) was used if the size of the expected number was smaller than 5. The two-group comparison was performed using the Student t-test for independent variables or the U Mann-Whitney test according to the data distribution (or after data logarithmic normalization). The oxycodone dose time analysis was done based on the repeated measures ANOVA with a post-hoc contrast analysis. The data were logarithm-transformed due to the deviation from the normal distribution. The median oxycodone levels in the study groups are presented as time profiles (values in the following time points of observation) with distance-weighted least-squares smoothing. Area under curve (AUC) of oxycodone dose was calculated with the trapezoidal rule. Comparison between groups was done with the Student t-test for independent variables after data log-normalization. The time analysis of the NRS was performed based on the longitudinal mixed-model rank analysis with the analysis of contrasts, with the Benjamini-Hochberg correction for multiple comparisons. Factors related to adverse events were determined based on a multivariable (backward stepwise) logistic regression, while factors influencing the length of stay were assessed based on multivariable (backward stepwise) linear regression. Parameters were considered statistically significant with *p* < 0.05. The following software was used for calculations: Statistica 13.0 (TIBCO Inc., Palo Alto, CA, U.S.) Polish version, MS Office Excel.

## 3. Results

A total of 109 patients scheduled for elective total posterolateral hip replacement were included in the study. Three patients were not randomised due to the need for conversion from spinal to general anaesthesia. Another six patients failed to complete the study due to their inability to effectively use the PCA pump (*n* = 4) and the lack of optimal anatomical conditions for FICB (*n* = 2). Ultimately, 100 patients were included in the analysis (50 in the control group and 50 in the FICB group; Figure 1).

The mean age of patients was 65 years, with hypertension and overweight being the main comorbidities. Other comorbidities included ischaemic heart disease and diabetes, with no statistically significant differences between the study groups. There were statistically significantly more ASA III patients in the FICB group. The groups did not differ in terms of gender distribution or the level of the maximum experienced pain assessed in the preoperative period (Table 1).

### 3.1. Pain Score

Statistically significantly lower resting NRS scores were found in the FICB group (*p* < 0.001) at all time points except for 48 h (Figure 2). Also, pain intensity was significantly lower in the FICB group during rehabilitation on day 1 (NRS 5 (4–5) vs. 6 (5–6), respectively; *p* < 0.001) and 2 postoperatively (NRS 5 (4–5) vs. 5 (5–6), respectively; *p* < 0.01) Figure 3. In the control group, there was a statistically significant decrease in the score over time (*p* < 0.01), as opposed to the FICB group (*p* = 0.28).

### 3.2. Opioid Consumption

The total consumed dose of oxycodone was statistically significantly higher in the control group than in the FICB group (*p* < 0.001). The total administered dose of oxycodone was 61.4 ± 15.8 mg in controls and 40.0 ± 10.2 mg in the FICB group (Table 1). The AUC of oxycodone dose was also higher in controls than in the FICB group (Table 1, Figure 4). The time profiles for oxycodone doses, which are the median oxycodone levels over time in the study groups, are shown in Figure 5. We also found statistically significantly longer time to the first analgesic intervention in the FICB group compared to controls (4.5 h ± 1.0 h vs. 3.0 h ± 0.7 h; *p* < 0.001)—Figure 6.

### 3.3. Adverse Events

We found significant differences between the groups in terms of adverse events monitored during the postoperative period. Postoperative nausea and vomiting (*p* < 0.05 and *p* < 0.001, respectively), episodes of bradycardia (*p* < 0.001), and hypotension (*p* < 0.01) were significantly less common in the FICB group (Table 2).

The following factors were included in the multivariable analysis: group, gender, age, BMI, occurrence of HA, DM, CAD, NRS scale, ASA scale (III vs. II), the use of ephedrine and noradrenaline.

For nausea and bradycardia, only the total dose of oxycodone proved to be a risk factor (OR with 95% CI: 1.098 (1.042–1.157), *p* < 0.001 and 1.069 (1.038–1.100), *p* < 0.001; respectively). FICB and hypertension were factors reducing the risk of hypotension (OR with 95% CI: 0.146 (0.039–0.552), *p* < 0.01 and 0.203 (0.061–0.674), *p* < 0.01; respectively).

### 3.4. Time to Discharge

A statistically significant difference was found between the groups for the time to discharge. The LOS was longer in controls by an average of 1.1 day [95% CI: 0.7–1.4] compared to FICB patients (6.6 ± 0.7 vs. 5.6 ± 0.9) (Table 1).

### 3.5. Satisfaction with the Analgesic Treatment Used

We found differences in patients’ satisfaction with the analgesic treatment used between the groups. A total of 80% of FICB patients were very satisfied or satisfied with the postoperative analgesia compared to only 30% in the control group. No patients were reporting to be ‘very dissatisfied’ with analgesia in the FICB group compared to 6% in the control group.

## 4. Discussion

Although almost a decade has passed since the first description of FICB [15], reports on its applicability for elective total posterolateral hip replacement are still sparse. Furthermore, most of the available studies and meta-analyses concern mainly trauma patients, in whom the use of FICB reduces pain intensity, the need for opioids, and the rates of complications arising from their systemic use [22,23,24].

In the case of patients qualified for elective THR, FICB was found to significantly reduce the need for postoperative opioids [25,26]. Desmet et al. showed that the block effect in the form of reduced need for opioids was, however, demonstrated for the population of patients after anterior THR [25]. Reports on the efficacy of this method in the posterolateral approach are missing. Our analysis in the group of FICB patients showed a significantly lower level of postoperative pain both at rest and during rehabilitation. It should be noted that the median pain intensity measured with the NRS did not exceed a score of 4 at rest at any measurement time point, which proves its high analgesic efficacy. An assessment of pain intensity during rehabilitation in the non-FICB group showed the median NRS score of 6 (after 24 h) and 5 (after 48 h), which indicates unsatisfactory control of nociceptive experiences. The median NRS score for the same parameter was statistically lower in the FICB group and was 5 for both measurements.

Although the use of opioids has been an unquestionable state of the art in anesthesia and postoperative pain management, there is emerging evidence that it can be associated with many sides effects not only in the immediate intra- and postoperative period (like respiratory depression, bradycardia, hypotension, PONV), but also affects long-term outcomes and influences the entire life of patients, like potentially developing opioid dependence or opioid-induced hyperalgesia [27,28]. Therefore, it is extremely important to take all measures to reduce the use of opioids in the perioperative period. Our study showed higher consumption of opioids in controls (mean total dose of oxycodone during 48 h in this group was 61.4 mg vs. 40.0 mg in the FICB group). The groups also differed in the time to first analgesic intervention. Patients in the FICB group needed the first rescue opioid dose on average 4.5 h after the surgery. This time was 90 min shorter in controls. These results are consistent with the meta-analysis by Liu et al. and Gao et al., where FICB was found to reduce the need for opioids and pain intensity in patients undergoing hip replacement [22,29]. Similar findings were presented by Desmet et al., who assessed the efficacy of FICB in anterior THR [25].

Postoperative nausea and vomiting, reduced gastrointestinal motility, and excessive somnolence are the main postoperative adverse effects related to systemic opioids. These adverse effects were significantly more common in the control group due to the increased use of oxycodone. Similar to our study, Xiao-yan Zhang and Gao Yun showed a similar difference in the incidence of opioid-induced adverse effects in their meta-analyses, which included 372 and 325 patients undergoing THR, respectively [26,29].

Discharge after hip replacement depends on the fulfillment of several key criteria. It is recommended that patients may be discharged if they are mentally and physically able to continue rehabilitation, move about, and travel over short distances [30]. Our analysis showed that functional independence and meeting the discharge criteria were achieved significantly earlier by FICB patients. Their length of stay was shorter by a mean of 1 day compared to controls. Zhang et al. showed similar results in their meta-analysis, where FICB reduced the length of stay [26].

Among the many factors affecting patient’s satisfaction with the treatment process, the degree of control of nociceptive experiences in the perioperative period is a key factor [31,32]. An analysis of data included in a paper assessing postoperative satisfaction in patients hospitalised at the New England Medical Center in Boston showed that 90% of patients undergoing THR were satisfied or very satisfied with the analgesic treatment [32]. In our FICB group, we obtained similar results (80% of FICB patients were satisfied or very satisfied), which confirms the use of an optimal analgesic protocol in this group of patients. It should be noted that only 30% of controls were satisfied or very satisfied with the analgesia used. At the same time, dissatisfied patients accounted for the majority (36%). These results do not allow for the routine recommendation of the analgesic treatment implemented in this group.

There are some limitations to our study. First of all, the sample size was small and the results should be confirmed in a large study. Also, further studies should be designed as multi-centre studies to exclude local factors affecting the final results. Another limitation is restricting the time of follow-up to 48 h without taking into account the long-term observation. Extending the follow-up to 1, 3, and 6 months would allow for observing the development of chronic pain in individual groups. Furthermore, the use of various adjuvants, which prolong the action of the local anesthetics used, could improve the results obtained. Open label design of the study was also a significant study limitation. Patients and assessor blinded design of the study would probably be more effective in reducing the risk of potential responder bias. Another imitation was the lack of a uniform patient sedation protocol (propofol infusion or headphones with relaxing music) that would translate into change in postoperative pain.

## 5. Conclusions

Compared to intravenous multimodal analgesia, supra-inguinal fascia iliaca compartment block (S-FICB) is an effective form of analgesia for elective posterolateral total hip replacement. It reduces the need for opioids, and thus the rates of complications resulting from their systemic administration. It also reduces the length of hospital stay and ensures a high level of patient satisfaction with the analgesic treatment used. The obtained results allow for recommending S-FICB for an elective posterolateral hip replacement as an effective and safe method of postoperative analgesia.

## Figures and Tables

**Figure 1 ijerph-18-04891-f001:**
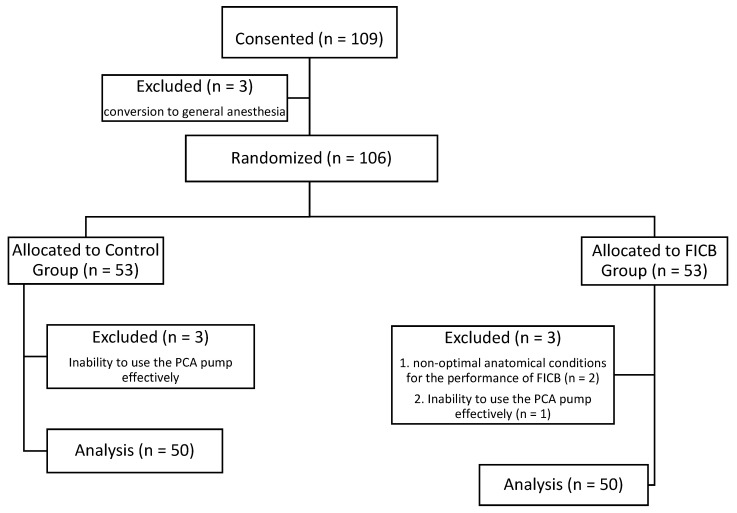
Participant flow diagram.

**Figure 2 ijerph-18-04891-f002:**
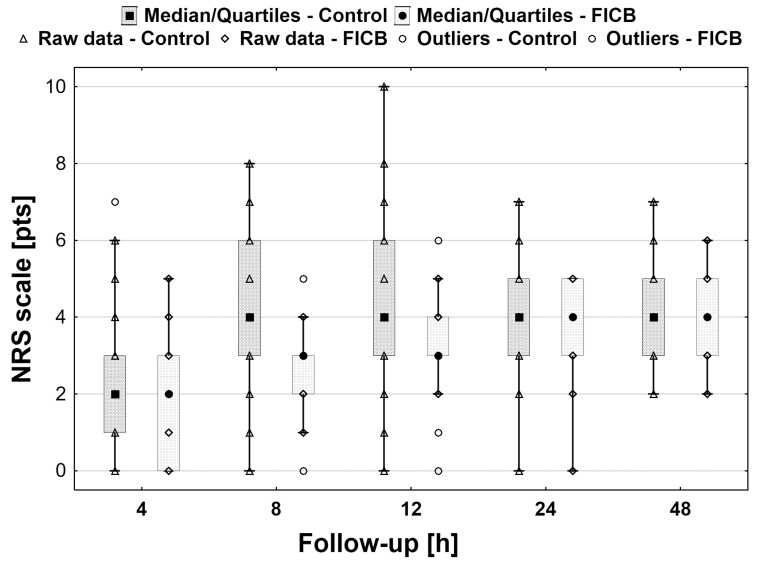
Box-plot of resting NRS in the control and FICB group. Significantly lower resting NRS scores were found in the FICB group at all time points except for 48 h.

**Figure 3 ijerph-18-04891-f003:**
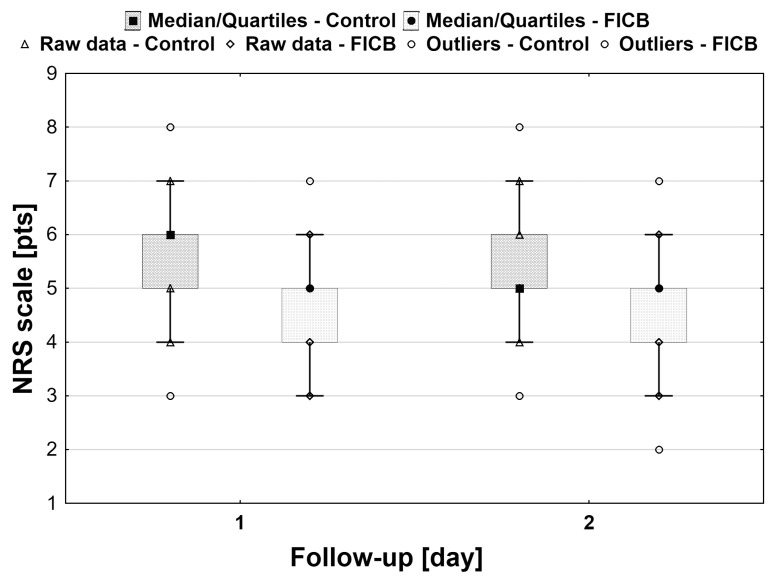
Box-plot of NRS in the control and FICB group during rehabilitation. Pain intensity was significantly lower in the FICB group during rehabilitation on day 1 and 2 postoperatively.

**Figure 4 ijerph-18-04891-f004:**
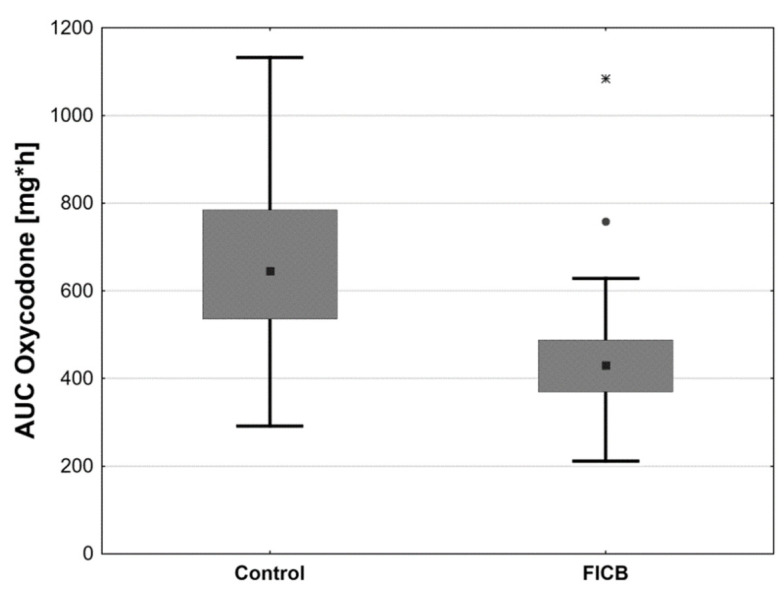
Box plot of oxycodone AUC in the control and FICB group.

**Figure 5 ijerph-18-04891-f005:**
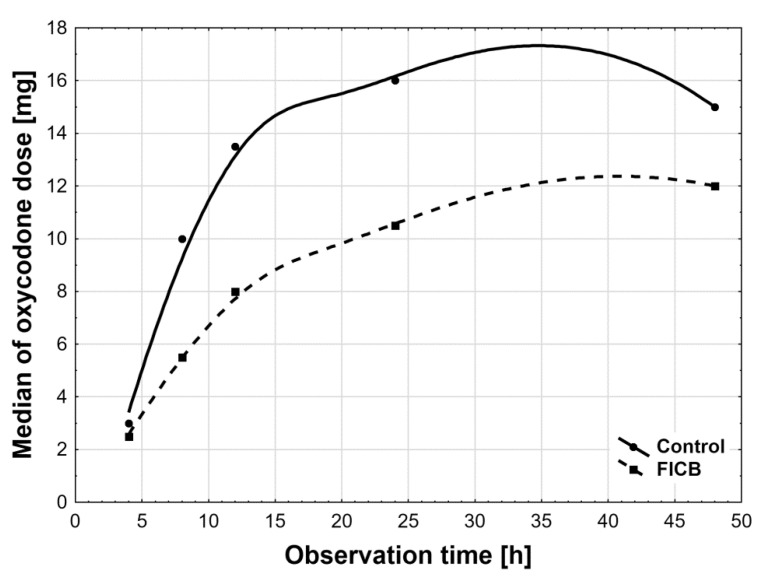
The median oxycodone levels over time of the study groups (distance-weighted least squares smoothing).

**Figure 6 ijerph-18-04891-f006:**
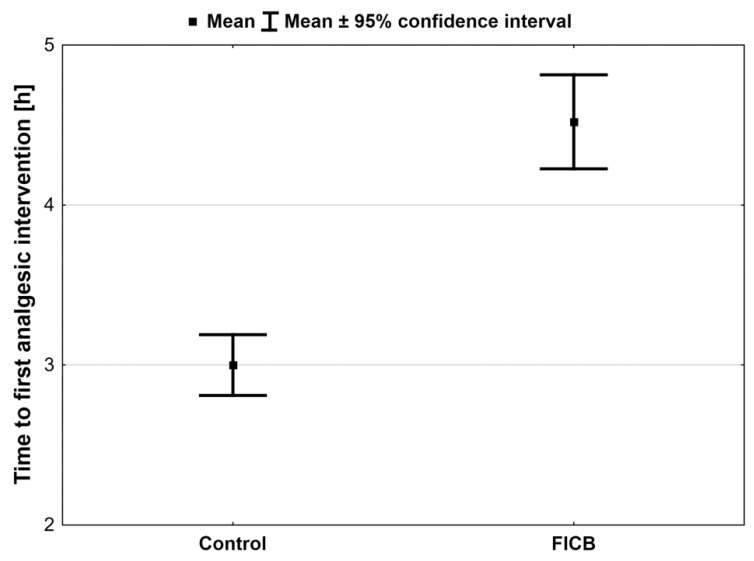
Time to the first analgesic intervention. Significantly longer time to the first analgesic intervention in the FICB group compared to controls.

**Table 1 ijerph-18-04891-t001:** Comparison of the evaluated descriptive parameters between the study groups.

	Controls *n* = 50	FICB *n* = 50	*p*
Female sex [N(%)]	28 (56.0)	29 (58.0)	0.84
Age [years]	65 ± 9	65 ± 12	0.89
BMI [kg/m^2^]	28.1 ± 2.9	27.0 ± 3.0	0.06
HA [N(%)]	37 (74.0)	29 (58.0)	0.09
DM [N(%)]	11 (22.0)	9 (18.0)	0.62
IHD [N(%)]	9 (18.0)	6 (12.0)	0.40
Overweight [N(%)]	27 (54.0)	30 (60.0)	0.63
NRS before	6 (5–8)	5 (4–7)	0.07
ASA II/III [N(%)]	34/16 (68/32)	23/27 (46/54)	<0.05
LOS [days]	6.6 ± 0.7	5.6 ± 0.9	<0.001
Σ Oxycodone dose PCA [mg]	61.4 ± 15.8	40.0 ± 10.2	<0.001
Oxycodone dose over time [mg * h]	646 (536–784)	430 (370–488)	<0.001

mean ± standard deviation or median (lower quartile–upper quartile); BMI, Body Mass Index; HA, Hypertonia Arterialis; DM, Diabetes Mellitus; IHD, Ischaemic Heart Disease; NRS, Numerical Rating Scale; LOS, length of stay in the hospital.

**Table 2 ijerph-18-04891-t002:** An inter-group comparison of the incidence of postoperative adverse events.

	Controls	FICB	*p*
Nausea	47 (94.0)	32 (64.0)	<0.05
Vomiting	26 (52.0)	9 (18.0)	<0.001
Hypotension	14 (28.0)	4 (8.0)	<0.01
Bradycardia	16 (32.0)	2 (4.0)	<0.001
Oversedation	12 (24.0)	9 (18.0)	0.46
Constipation	9 (18.0)	4 (8.0)	0.14

## Data Availability

The datasets used and/or analyzed during the current study are available from the corresponding author upon reasonable request.

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
