# Peer review of "Effectiveness of Fascia Iliaca Compartment Block after Elective Total Hip Replacement: A Prospective, Randomized, Controlled Study"

_ijerph, 2021, doi:10.3390/ijerph18094891_

Round 1
Reviewer 1 Report
Dear Authors,
May I ask you to address all my concerns, which you can find below?
General comments:
- Why did you choose open label design as it is usually dedicated to compare very similar treatments/products? Single blinded or patients and assessor blinded design would probably be more appropriate and effective in reducing the risk of bias. Please comment on this. Additionally, open label design should be indicated as a significant study limitation. This should be meticulously discussed in the Limitations section of the manuscript.
Specific comments:
- page 2, line 75: What was the rationale for establishing the cut-off value of 75 years and BMI of 19 - 30 as the inclusion/exclusion criteria?
- page 3, line 116 - 119: Please specify the term 'pressure drop below 70 mmHg'. Be more specific, is it MAP or SBP?
- page 4, line 163 - 173: Please provide detailed description regarding outcomes data collection and assessment/measurement. Who collected the data. Was she or he aware of trial allocations? If not, potential bias should be discussed in Limitations section.
- page 4, line 174 - 198: sample size calculation is missing. It should be discussed. Probably the trial is underpowered and this should be indicated in Limitations as well.
- page 5, line 205 - 211: Do not duplicate demographic data in the text and Table 1.
- page 9, line 258 - 263: Information regarding patients satisfaction seems to be not relevant as all patients were aware of trial allocation - significant risk of bias. It should be indicated as another significant trial limitation.
- Limitations: this section should be updated and rebuild in line with comments above.
- Conclusions: not fully supported by results. You mentioned that compared to intravenous multimodal analgesia, S-FICB is an effective form of analgesia for elective postero-lateral total hip replacement. However, in the trial design you tested S-FICB plus intravenous analgesia against intravenous analgesia only. So you can only conclude that S-FICB is an effective adjunct to standard analgesia, not solely independent mode of analgesia.
Author Response
Dear Reviewer,
First of all, thank you for the effort you put into reading the manuscript as well as for valuable review comments.
Please find attached our reply to the Review Raport below.

Reviewer 2 Report
Major comments
- Lines 57-58 make that statement that regional blocks are controversial without further explanation. The introduction would benefit from a clear statement of the problem that the study is aiming to solve which seems to be the lack of clinical adoption of regional blocks for hip replacement. Without this statement, it’s unclear why this study is needed.
- No mention of potential responder bias is included. Patients allocated to the control group would know that they did not receive the additional procedure and may therefore report higher pain scores. This would not be present if both groups underwent the procedure but the control group received a saline infusion instead of drug. Since all outcomes were based on patient’s experience/reports of pain, this bias affects all outcomes reported.
- Lines 109-111: were there differences in outcomes based on whether patients received Propofol or headphones? This is a significant difference in sedation which may affect patient’s pain experience in the postoperative period.
- Statistical analysis section does not describe the AUC analysis done for figures 3 and 4
- The number of patients approached, refused and consented are not provided.
- Table 1: define sum Oxy dose and oxy dose (mg * h). these terms/calculations are not defined in the table or in the methods.
- Figure 2: The mean/median? values of postoperative pain ratings in the control and FICB groups look to be very similar so it’s difficult to see how any difference between these groups is statistically significant.
- Lines 218-220: Figures should be provided to show the data presented in these lines so the reader can see the full distribution of the data.
- Figure 3: the graph doesn’t match the results as stated in the text.
- Figure 4: The transformation of the oxycodone dosage makes interpreting these findings difficult. Either refrain from using the transformation or explicitly interpret these findings in the text.
- Figure 4-5: “time profile” needs to be clarified – and preferably replaced with a better word as it’s unclear what the authors mean with this term.
- Figure 5: The figure shows curves yet there’s no description in the methods for how the curve was constructed.
- Figure 6: Consider representing these data using a Kaplan-Meier curve
- Lines 254-257: use linear regression for LOS to show that FICB is an independent predictor after controlling for the other outcome variables.
- Lines 278-280: revise sentence- the non-FICB group’s NRS was just one point higher than FICB which does not indicate “unsatisfactory control of nociceptive experiences”. Both are moderate levels of pain which may not be clinically distinguishable.
Minor comments
- Lines 44-45: how is “longer professional activity” relevant to an increased number of procedures?
- Lines 52-57: Revise sentence – perhaps break it up into 2 sentences as it doesn’t make sense as written
- Line 65: change “primary endpoint” to “purpose”. The primary endpoint of the study is not to assess
- Line 76: define “ASA physical status”
- Line 96: Clarify Lyrica dose before procedure- lyrica was given PO at “75 mg an hour”?
- Lines 104-105: how was an aerosol disinfectant used to assess the extent of the sensory block
- Line 178: change “qualitative data” to categorical or other appropriate descriptor
- Figure 1 – small font is difficult to read
- Table 1 – include in the table footer the definitions for abbreviations for terms in the table
- Figure 2: the scale on the y-axis should be 0-10 not -2 to 12
- Figure 2: both sets of data should be plotted on the same graph to allow for better comparison between the groups.
- Lines 224-230: the methods needs to include a description of how opioid consumption was calculated as the data are presented in 4 hour intervals, but it’s likely that opioids were not consumed in this same manner.
- Lines 250-253: show full final regression model and include appropriate interpretation of the independent variables in the text.
Author Response

(The authors gave the same response as above.)

Round 2
Reviewer 1 Report
Dear Authors,
Thank you very much for addressing all my concerns. I have no additional questions.
Author Response
Dear Reviewer, thank you for your thorough evaluation and valuable comments, as well as a positive review of the manuscript.
Reviewer 2 Report
While some changes have been made to the manuscript, the authors have largely remained unresponsive to my primary concerns regarding the manuscript. This is especially true of questions and concerns I have as a reader of this manuscript about their purpose, methods, data, and interpretation. As a scientist myself, I aim to do what is needed to ensure my work is understood and interpreted appropriately by the reader. Many of my suggestions are to aid future readers in understanding this work and using these results as appropriate in their field. The reluctance of the authors to address many of these queries, leaves many ambiguities and concerns about the integrity of their data which is unfortunate.
Recommendations:
Lines 52-63:
Despite revising the introduction, the purpose of the study is not clear and is not congruent with the problem presented. The problem is that “the choice of an optimal form of analgesia… is not defined” and there are no uniform guidelines. The primary and secondary endpoints is to assess FICB efficacy and analgesic use. The introduction would benefit from a purpose statement which addresses the problem rather than presenting a series of endpoints.
The authors did not address potential responder bias in the paper. It is clear that a true comparison group was not feasible. However, this needs to be explicitly stated as a limitation of the study.
I appreciate that the author’s believe that no differences between Propofol and music would translate into change in postoperative pain. However, it is appropriate to include this comparison so the reader is assured that an anesthetic drug given during the procedure did not interfere with postprocedural pain ratings.
Figure 2 – because the differences in groups is not apparent, presentation of the individual points is needed to clarify how statistical significance may be achieved in this analysis. In addition, the plotting of the data is inefficient. Time should be plotted on the x axis with the 2 groups as categories so that both groups are plotted side by side at each time point.
Figure 3 – individual points should be plotted in addition to the box to show the distribution of values. The plotting is also inefficient and time should be plotted on the x axis as described in the comments for Figure 2.
Despite its usage in scientific works, the log transformation and use of “time profile” makes interpreting this data difficult for the reader.
Use different wording for “longer professional activity” as this is difficult for the reader to understand without further explanation.
How is Lyrica give PO 75mg/hour? Was each patient given a 75mg tablet each hour? I appreciate the reference, but this dosage doesn’t make sense.
Qualitative data are not nominal or ordinal – I recommend omitting “qualitative” and just stating nominal/categorical/ordinal data as appropriate. These are all quantitative data.
To provide full results, please show the full final regression models and include appropriate interpretation of the independent variables in the text.
Author Response
Dear Reviewer,
First of all, thank you for the effort you put into reading the manuscript as well as for valuable review comments.
Please find attached our reply to the Review Raport.
